# Antineutrophil Cytoplasmic Antibody-Associated Vasculitis and the Risk of Developing Incidental Tuberculosis: A Population-Based Cohort Study

**DOI:** 10.3390/medicina59111920

**Published:** 2023-10-30

**Authors:** Shan-Ho Chan, Ming-Feng Li, Shih-Hsiang Ou, Mei-Chen Lin, Jen-Hung Wang, Po-Tsang Lee, Hsin-Yu Chen

**Affiliations:** 1Department of Medical Imaging and Radiology, Shu-Zen Junior College of Medicine and Management, Kaohsiung 82144, Taiwan; shchan@ms.szmc.edu.tw (S.-H.C.); oudinot@gmail.com (M.-F.L.); 2Department of Radiology, Kaohsiung Veteran General Hospital, Kaohsiung 813414, Taiwan; 3Division of Nephrology, Department of Internal Medicine, Kaohsiung Veterans General Hospital, Kaohsiung 813414, Taiwan; blueyeou1104@gmail.com (S.-H.O.); ptlee@vghks.gov.tw (P.-T.L.); 4School of Nursing, Meiho University, Pingtung 91202, Taiwan; 5Management Office for Health Data, China Medical University Hospital, Taichung 404327, Taiwan; coolindm@gmail.com; 6College of Medicine, China Medical University, Taichung 40402, Taiwan; 7Department of Medical Research, Hualien Buddhist Tzu-Chi General Hospital, Hualien 970473, Taiwan; jenhungwang2011@gmail.com

**Keywords:** antineutrophil cytoplasmic antibody-associated vasculitis, tuberculosis, population-based study

## Abstract

*Background and Objectives*: Treatment for antineutrophil cytoplasmic antibody-associated vasculitis (AAV) must deal with immunosuppression, as well as infections associated with a compromised immune system, such as tuberculosis (TB). Our aim was to examine the risk of incidental TB after diagnosis of AAV. *Materials and Methods*: This retrospective population-based cohort study was based on the data from the National Health Insurance Research Database in Taiwan. Patients with newly diagnosed granulomatous polyangiitis or microscopic polyangiitis were identified between 1 January 2000 and 31 December 2012. The primary outcome was risk of incidental TB. Cox proportional hazard models were used to evaluate the association between AAV and incidental TB. *Results*: A total of 2257 patients with AAV and a propensity-score matched cohort of 9028 patients were studied. Overall, patients with AAV were at a 1.48× higher risk of contracting incidental TB than the patients in the matched cohort (adjusted HR 1.48; 95% confidence interval [CI], 1.02–2.15). Note that the highest risk of contracting incidental TB was in the first two years following a diagnosis of AAV, with a nearly 1-fold increase in risk (adjusted HR, 1.91; 95% CI, 1.01–3.60). Female AAV patients were 3.24× more likely than females without AAV to develop TB (adjusted HR 3.24; 95% CI, 1.85–5.67). *Conclusions*: Patients with AAV exhibit a 48% elevated TB risk, notably, a 91% increase within the first two years postdiagnosis. Female AAV patients face a 3.24 times higher TB risk compared to females without AAV. This study is limited by potential misclassification and overestimation of AAV cases. Clinicians should closely monitor TB risk in AAV patients, especially in females and the initial two years following diagnosis.

## 1. Introduction

Immunosuppressive treatment for antineutrophil cytoplasmic antibody (ANCA)-associated vasculitides (AAV) must deal infection associated with a compromised immune system, such as tuberculosis (TB) in TB endemic areas. TB is a major health threat worldwide [1], with roughly 9 million diagnoses and 1.5 million deaths annually, based on estimates by the World Health Organization (WHO) [2]. TB is endemic in Taiwan. In 2019, the annual incidence was 37 per 100,000 persons, and the annual mortality was 2.3 per 100,000 [3].

AAV patients face a high risk of infection. In a 325-patient AAV cohort with 2307 patient-years of follow-up, 40% had at least one severe infection. The incidence rate was 9.1/100 patient-years, peaking in the first year postdiagnosis [4]. In European Vasculitis Society trials, infection led to the hospitalization of 30% of the AAV patients, and was the leading cause of death [5]. Another report on 489 AAV patients determined that 42% of the instances of infection were pulmonary in nature [6]. In TB endemic areas, understanding the risk of incidental TB following AAV diagnosis is crucial. However, due to the relatively low prevalence of AAV and TB, research exploring AAV and subsequent TB risk is lacking.

Our study aims to assess the risk of incidental TB following AAV diagnosis. Using a novel algorithm [7], we identified patients with granulomatosis with polyangiitis (GPA) and microscopic polyangiitis (MPA) from a national database to evaluate their subsequent TB risk. 

## 2. Materials and Methods

### 2.1. Data Source

The Taiwan National Health Insurance (NHI) has covered more than 97% of the residents of Taiwan since 1996. The NHI Research Database (NHIRD) provides an enormous representative cohort with a long period of follow-up for the epidemiological analysis of rare diseases. Deidentified secondary data contains all registry and administrative claims information, ranging from demographic data to details pertaining to ambulatory and inpatient care. This study assembled nationwide hospitalization files based on the NHIRD. All historical diagnoses in the database were coded in accordance with the International Classification of Disease, Ninth Revision, Clinical Modification (ICD-9-CM). Patient consent was waived due to the retrospective nature of this study, and participant data was be deidentified to ensure anonymity. The Research Ethics Committee of China Medical University and Hospital in Taiwan approved this study (CMUH-104-REC2-115-(AR4)). 

### 2.2. Study Design

Our objective in this population-based, observational, retrospective cohort study was to characterize the association between AAV patients and incidental TB. We first identified all adult subjects (≥20 years) enrolled in 2000, and then extracted all relevant data pertaining to those subjects throughout the study period of January 2000 to December 2012. The resulting dataset included demographic characteristics, diagnosis and procedure codes, drug prescriptions, comorbidities, and information about outpatient visits and hospital admissions. 

### 2.3. Selection of Patients with ANCA-Associated Vasculitis

All participants underwent a 12-month burn-in period. Individuals with any of the following diagnoses during the burn-in period were excluded: (1) GPA or MPA diagnosis; (2) hepatitis B (ICD-9 CM: 070.2, 070.3, and V02.61) or hepatitis C (ICD-9 CM: 070.41, 070.44, 070.51, 070.54, and V02.62); (3) diagnosis of renal failure (ICD-9 CM: 639.3, 586, V56.8, and V45.1), glomerulonephritis (ICD-9 CM: 580–582, and 583.1–583.4); (4) hemoptysis (ICD-9 CM: 786.3). 

The identify of GPA cohort was made by individuals aged ≥ 20 years with medical claims of GPA (ICD-9-CM diagnosis code 446.4), and at least two outpatient visits or one inpatient visit between 1 January 2000 and 31 December 2012. 

The identify of MPA cohort was made by individuals aged ≥ 20 years with medical claims of MPA (ICD-9-CM diagnosis code 447.6), and at least two outpatient visits or one inpatient visit between 1 January 2000 and 31 December 2012.

Ultimately, the AAV cohort comprises both the GPA and MPA cohorts. The exclusion criteria were as follows: (1) history of tuberculosis prior to diagnosis of AAV; (2) missing basic information; (3) age < 20 years old.

### 2.4. Matched Cohort Selection

Propensity score matching was used to reduce bias in patient selection, and generate matched pairs of patients, thereby making it possible to compare the outcomes of AAV patients and the matched cohort [8]. To further consider the robustness of the correction, we applied 1:4 nearest propensity score matching (PSM) to adjust for the variables listed in Table 1, and compared the results to the multivariate Cox model. PSM via the nearest-neighbor method, initially to the eighth digit, and then as required to the first digit, was used in this study. Matches were first made within a caliper width of 0.0000001, and then the caliper width was increased for unmatched cases to 0.1. We reconsidered the matching criteria and performed a rematch according to the greedy algorithm, including the following baseline variables: year of index date, age, sex, monthly income, urbanization, CCI score, and comorbidities (diabetes, hypertension, hyperlipidemia, atrial fibrillation, valvular heart disease, parkinsonism, and autoimmune disease). A standardized mean difference of < 0.1 indicated a negligible difference in the distribution between both cohorts [9,10].

### 2.5. Variables and Comorbidity

Baseline demographic characteristics included age, sex, monthly income, and urbanization level of the patients’ places of residence. The health status of patients was assessed systematically using the Charlson Comorbidity Index (CCI). Each increase in the CCI represents a stepwise increase in cumulative mortality. A score of 0 corresponds to a 99% 10-year survival rate, whereas a score of 5 corresponds to a 34% 10-year survival rate [9]. Instances of comorbidity were designated by at least two outpatient medical claims or one inpatient medical claim of diabetes mellitus (ICD-9-CM: 250, A181), hypertension (ICD-9-CM: 401–405, A260, A269), hyperlipidemia (ICD-9-CM: 272.0–272.4), atrial fibrillation (ICD-9-CM: 427.31), valvular heart disease (ICD-9-CM: 390–398, 424), parkinsonism (ICD-9-CM: 332, A221), or autoimmune disease (ICD-9-CM: 710, 714). 

### 2.6. Outcome Measures

The primary outcome was the risk of incidental TB after the index date. Patients who met the following two conditions were considered as having incidental TB: (1) at least two outpatient medical claims or one inpatient medical claim of TB (ICD-9-CM: 010–018, A02); (2) continuous prescriptions of antibiotics for the treatment of TB for at least 28 days (ATC code: J04A). 

### 2.7. Statistical Analysis

The distributions of age, gender, and comorbidities in the AAV cohort and matched control cohort were indicated by numbers and percentages. Differences between two cohorts were tested using the chi-square and *t*-test, respectively, for categorical and continuous variables. Among patients without event occurrence, the length of follow-up (in person-years) was calculated from the index date to either the date of diagnosis for cardiovascular disease, death, or the last follow-up prior to 31 December 2013. Hazard ratios (HRs) and 95% confidence intervals (95% CI) were estimated using the Cox proportional hazard models in order to evaluate the association between AAV and incidental TB. Schoenfeld residuals were used to evaluate assumptions pertaining to the Cox proportion. The link between AAV and incidental TB was also evaluated using stratification analysis based on age, gender, CCI, and comorbidities. The multiple Cox proportional hazard model was used to estimate HRs after adjusting for age, gender, and comorbidities. Survival curves in the two cohorts were plotted using the Kaplan–Meier method and tested using the Log-Rank test. Interaction tests were used to determine interactions between subgroups and the risk of incidental TB. In research using Taiwan’s NHIRD, missing data are typically excluded. However, the current study contains no missing data. All statistical analyses was performed using SAS statistical software, version 9.4 (SAS Institute Inc., Cary, NC, USA). The Kaplan–Meier plot was plotted using R software, version 4.3.1. Statistical significance was determined using two-tailed tests (*p* < 0.05). 

## 3. Results

### 3.1. Baseline Demographic Data

The study flowchart is in Figure 1. Our study population included 2257 and 9028 patients in the AAV and matched-cohorts, respectively. 

As shown in Table 1, there were no significant differences between AAV patients and the matched cohort in terms of baseline demographic data. The mean ages in the AAV group and matched cohort group were 54.6 years (standard deviation [SD] = 17.5) and 53.1 years (SD = 17.0), respectively.

### 3.2. Risk of Incidental TB in Entire Cohort

Table 2 shows that the risk of developing incidental TB was 1.48 times higher in the AAV cohort than in the matched cohort (adjusted HR: 1.48, 95% CI = 1.02–2.15). The risk of developing incidental TB was 2.49 times higher among male patients in the entire cohort than female patients in the entire cohort (adjusted HR: 2.49, 95% CI = 1.70–3.63). Compared to patients aged < 40 years, the risk of developing incidental TB was 2.75 times higher among patients aged 40–65 years old, and 9.10 times higher among those aged ≥ 65 years (adjusted HR: 2.75, 95% CI = 1.26–6.02; adjusted HR 9.10, 95% CI = 4.12–20.06, respectively). Compared to patients with 0 CCI score, the risk of developing incidental TB was 1.92 times higher among patients with CCI score ≥ 2 in the entire cohort (adjusted HR: 1.92, 95% CI = 1.19–3.10). Figure 2 presents the Kaplan–Meier analysis of cumulative incidence of TB, wherein the incidence of TB was higher in the AAV cohort than in the control group (Log-Rank *p* = 0.0661). Schoenfeld residuals obtained during the study revealed that the proportional hazard might not be against the assumption (*p* = 0.99) in each model. 

### 3.3. Risk of Incidental TB: Stratification and Interaction Tests

As shown in Table 3, the risk of developing incidental TB was 3.24 times higher among female AAV patients than among females in the matched cohort (adjusted HR: 3.24, 95% CI = 1.85–5.67), with a significant *p* value for interaction (*p* for interaction = 0.001). The risk of contracting incidental TB was similar between the two groups when stratified in terms of age, CCI score, diabetes, hypertension, or hyperlipidemia. There was no significant interaction by age, CCI score, diabetes, hypertension, or hyperlipidemia (age: *p* for interaction = 0.707; CCI score: *p* for interaction = 0.597; diabetes: *p* for interaction = 0.514; hypertension: *p* for interaction = 0.435; hyperlipidemia: *p* for interaction = 0.286). As shown in Table 4, AAV patients who continued follow-up for less than two years were at a 1.91 times higher risk of contracting incidental TB than were those in the matched cohort (adjusted HR: 1.91, 95% CI = 1.01–3.60, *p* = 0.046).

## 4. Discussion

This nationwide population-based study using propensity score matching provided strong evidence that the risk of contracting incidental TB was nearly 50% higher in AAV patients than those without AAV. Female AAV patients had nearly 3-fold higher risk of subsequent TB than females without AAV. When compared with the matched cohort, the risk of incidental TB was highest for the first 2 years after diagnosis of AAV, with nearly 1-fold increased risk. Clinicians should closely monitor subsequent TB risk in AAV patients, especially in females and the initial two years following diagnosis.

In our study cohort, patients with AAV are associated with increased nearly 50% risk of incidental TB, and with nearly 1-fold increased risk (adjusted HR, 1.91; 95% CI, 1.01–3.60) within the first 2 years after diagnosis of AAV. Current research predominantly focuses on the potential of TB to trigger AAV, or on the prevalence of P-ANCA and C-ANCA in TB patients [11,12]. However, there is a paucity of studies investigating the risk of subsequent incidental TB following a diagnosis of AAV. AAV is an inflammatory disease characterized by vascular inflammation, similar to systemic lupus erythematosus (SLE). In previous studies, SLE was linked to an elevated risk of developing incidental TB, compared to patients without SLE (OR = 4.6) [13]. The susceptibility of AAV patients to TB could perhaps be explained by dysregulation of T cell responses and medications administered for the treatment of AAV. There has been a lack of research examining the immunity association of TB and AAV. One mechanism involved in immunity to TB is the delayed induction of TB-specific, Foxp3+ regulatory T (Treg) cells [14]. AAV patients have been linked to elevated numbers of circulatory T follicular helper cells (Tfh) and T follicular regulatory cells (Tfr), as well as an elevated Tfh2/Tfh1 ratio [15]. It is possible that an imbalance in the Treg reaction may play a role in the development of TB in AAV patients. According to the payment system of the NHI in Taiwan, the first line treatment for AAV focuses on steroids and cyclophosphamide. Rituximab is used in specific situations: (1) the patient fails to respond adequately to cyclophosphamide treatment over a period of four weeks following the onset of AAV; (2) the patient undergoes a recurrence of AAV after cyclophosphamide treatment; (3) the patient shows intolerance to cyclophosphamide. According to a population-based nested case-control study of nearly 6000 TB patients in Taiwan, the current, recent, past, ever, and chronic use of corticosteroids were all associated with an elevated risk of developing incidental TB [16]. In a population-based nested case-control study in Canada, the current use of disease modifying antirheumatic drugs, including cyclophosphamide, azathioprine, and cyclosporin, was associated with an elevated risk of developing incidental TB (adjusted OR = 23) [17]. The heavy burden imposed by immunosuppressant use following a diagnosis of AAV may explain the high risk of developing incidental TB within the first two years. Thus, we recommend the regular screening of AAV patients for incidental TB in endemic areas. 

In the current study, the risk of developing incidental TB was nearly 2-fold higher among female AAV patients than among females in the matched cohort (adjusted HR 3.24; 95% CI, 1.85–5.67; *p* < 0.001). A number of studies have reported on the risk of infection among female AAV patients. Female sex was identified as a significant predictor of infection in 1-year (n = 421) and 2-year studies (n = 374) of AAV patients [4,5]. However, a population-based study on 186 AAV patients in Sweden reported that sex was not associated with the risk of severe infection [18]. We observed no sex differences pertaining to the imbalance of Treg regulation in AAV patients. Thus, there is a need for a well-designed study aimed at exploring the mechanism underlying the susceptibility of female AAV patients to TB. 

This was the first study to explore the association between AAV and incidental TB, and our use of the Raimundo algorithm made it possible to overcome the limitations of ICD-9 coding, as they pertain to AAV [7]. Our results indicate that physicians in TB endemic areas should remain vigilant to the threat of incidental TB when treating AAV patients, particularly within the first two years. 

Our study has five major limitations. First, there was a risk of misclassification in terms of GPA and MPA diagnosis, due to the lack of an ICD-9-CM diagnosis codes specific to MPA. The accuracy of our MPA diagnosis algorithm relied on the manifestations of severe MPA and the integrity of administrative claim data. Alveolar hemorrhage was not included in this algorithm due to a lack of specific ICD-9-CM codes. We included hemoptysis as an alternative; however, this no doubt skewed the results. There was also the possibility that the administrative claim data obtained from the NHIRD were flawed, due to incomplete coding or misclassification. Nonetheless, we did not compare the outcomes of GPA and MPA separately; therefore, the effects of misclassification would no doubt be very small. Second, we identified far fewer cases of GPA than MPA, and we suspect that a portion of the GPA patients was misclassified as MPA. In this study, the AAV diagnostic criteria included criteria from the American College of Rheumatology (ACR), Chapel Hill Consensus Conference criteria (CHCC), and The European Medicines Agency (EMA) algorithm. None of these schemes reliably differentiate between GPA and MPA. Note that the critical pathological difference between GPA and MPA is the presence of granulomatous, which is easily missed due to sampling error. When using the CHCC approach, patients presenting with nasal disease or necrotizing vasculitis but no evidence of granulomatosis are labeled as MPA [19], thereby increasing the likelihood of a diagnosis of MPA. Nonetheless, the actual impact of misclassification is no doubt minimal. Third, the incidence of AAV in the current study was higher than in previous reports. In our overall cohort, the annual incidence of AAV was 13.90–35.83 per 100,000 patient years (Appendix A), whereas the AAV incidence reported in other studies showed far greater deviation. In Europe, the incidence of AAV has been estimated at 12.4, 15.16, 20.4, and 20.8 cases per million people in Germany, Spain, the United Kingdom, and Finland, respectively [20,21,22,23]. The incidence of AAV has been estimated at 33 cases per million people in the US, and 23 cases per million people in Argentina [24,25]. The reasons for this enormous range of variation can largely be attributed to diagnostic criteria. A longitudinal, retrospective, cohort study on kidney disease in Taiwan collected 6675 patients with pathologies of the kidney between January 2015 and December 2019. In that study, AAV was involved in 4.1% of the cases of primary glomerulonephritis [26]. The NHIRD used in the current study was a longitudinal cohort with 1,000,000 patients, and our study period was from 2000 to 2012. During that period, there were 53,839 cases of glomerulonephritis in the NHIRD; 4.1% of those cases equates to 2154 patients, which is close to the number of AAV cases in this study (2257). The algorithm developed by Raimundo et al. was meant to overcome limitations on ICD-9 coding in order to facilitate analysis of AAV with a rare disease. Nonetheless, it appears that this algorithm also increases the likelihood of overestimating the number of AAV cases. Large-scale, multicenter, randomized-controlled trials will be needed to overcome the limitations of administrative claims-oriented databases. Fourth, the algorithm employed for AAV case identification has not undergone validation, potentially introducing bias related to data accuracy. Fifth, this study does not account for the impact of immunosuppressive agents commonly used in AAV patients, such as steroids, cyclophosphamide, rituximab, azathioprine, or cyclosporine, on the risk of subsequent TB. Future developmental studies are needed.

## 5. Conclusions

Patients with AAV exhibit a 48% elevated TB risk, notably, a 91% increase within the first two years postdiagnosis. Female AAV patients face a 3.24 times higher TB risk compared to females without AAV. Clinicians should closely monitor TB risk in AAV patients, especially in females and the initial two years following diagnosis.

## Figures and Tables

**Figure 1 medicina-59-01920-f001:**
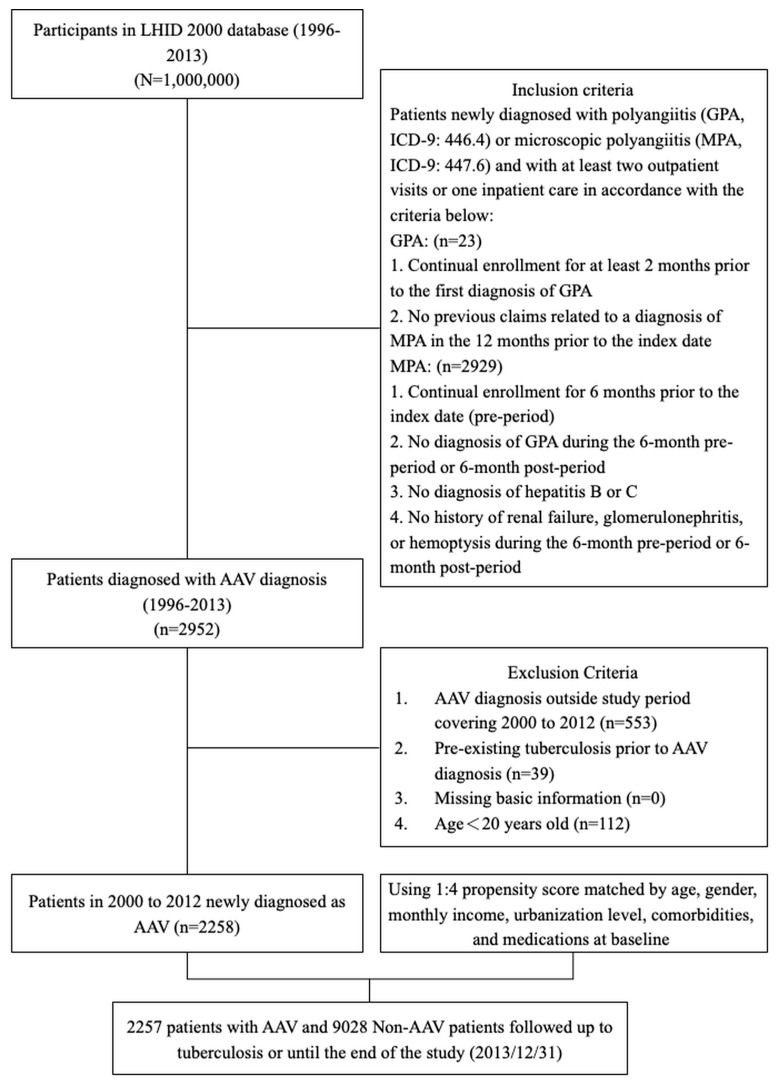
Study flowchart showing patient selection.

**Figure 2 medicina-59-01920-f002:**
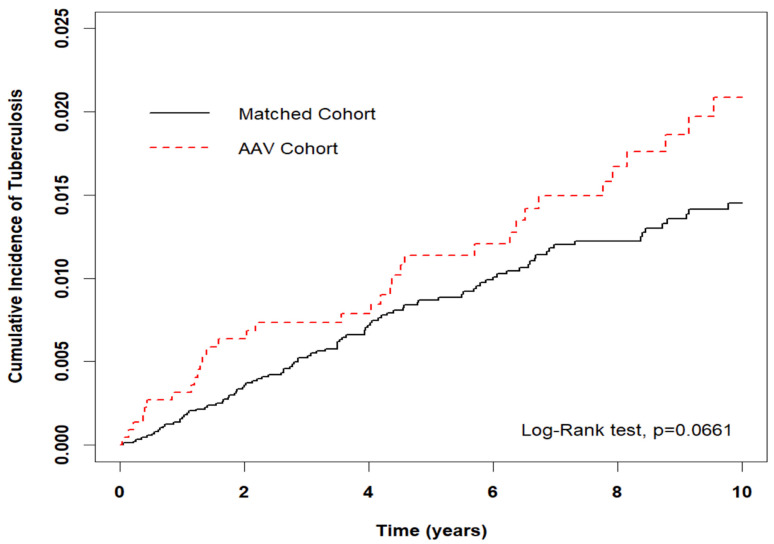
Cumulative incidence of tuberculosis in all patients with AAV and matched cohort.

**Table 1 medicina-59-01920-t001:** Demographic characteristics and comorbidities of AAV patients in Taiwan during 2000–2012.

Variable	AAV			Standardize Mean Difference (SMD) ^§^
	Total	No	Yes	
	n = 11,285	n = 9028	n = 2257	
	n	n (%)/Mean (SD)	n (%)/Mean (SD)	
Sex				0.030
Female	6028	4795 (53.1)	1233 (54.6)	
Male	5257	4233 (46.9)	1024 (45.4)	
Age at Baseline (years)				0.081
<40	2631	2055 (22.8)	576 (25.5)	
40–65	5261	4202 (46.5)	1059 (46.9)	
≥65	3393	2771 (30.7)	622 (27.6)	
Mean (SD)		54.6 (17.5)	53.1 (17.0)	0.086
Monthly Income (NTD)				0.072
0–15,840	4450	3540 (39.2)	910 (40.3)	
15,841–28,800	4939	3937 (43.6)	1002 (44.4)	
28,801–45,800	1389	1121 (12.4)	268 (11.9)	
>45,800	507	430 (4.8)	77 (3.4)	
Urbanization				0.081
1 (highest)	3540	2808 (31.1)	732 (32.4)	
2	3315	2693 (29.8)	622 (27.6)	
3	1954	1520 (16.8)	434 (19.2)	
4	2476	2007 (22.2)	469 (20.8)	
CCI Score				0.058
0	8840	7074 (78.4)	1766 (78.2)	
1	1258	981 (10.9)	277 (12.3)	
≥2	1187	973 (10.8)	214 (9.5)	
Baseline Comorbidity				
Diabetes	2396	1948 (21.6)	448 (19.8)	0.043
Hypertension	5031	4072 (45.1)	959 (42.5)	0.053
Hyperlipidemia	2882	2351 (26)	531 (23.5)	0.058
Atrial Fibrillation	230	189 (2.1)	41 (1.8)	0.020
Valvular heart disease	720	580 (6.4)	140 (6.2)	0.009
Parkinsonism	139	110 (1.2)	29 (1.3)	0.006
Autoimmune disease	239	179 (2)	60 (2.7)	0.045

^§^ A standardized mean difference of ≤0.1 indicates a negligible difference. Abbreviation: SD, standard deviation.

**Table 2 medicina-59-01920-t002:** Cox model measuring hazard ratio and 95% confidence intervals of TB patients with and without AAV.

Characteristics	No. of Events	Crude		Adjusted	
	(n = 142)	HR (95% CI)	*p*-Value	HR (95% CI)	*p*-Value
AAV					
No	104	Ref.		Ref.	
Yes	38	1.41 (0.98–2.05)	0.067	1.48 (1.02–2.15)	0.041
Sex					
Female	51	Ref.		Ref.	
Male	91	2.11 (1.50–2.98)	<0.001	2.49 (1.70–3.63)	<0.001
Age at Baseline (years)					
<40	8	Ref.		Ref.	
40–65	41	2.57 (1.20–5.47)	0.015	2.75 (1.26–6.02)	0.012
≥65	93	11.26 (5.47–23.19)	<0.001	9.10 (4.12–20.06)	<0.001
Monthly Income (NTD)					
0–15,840	65	Ref.		Ref.	
15,841–28,800	64	0.82 (0.58–1.16)	0.256	0.82 (0.57–1.20)	0.308
28,801–45,800	11	0.49 (0.26–0.93)	0.029	0.67 (0.35–1.31)	0.243
>45,800	2	0.24 (0.06–0.98)	0.047	0.34 (0.08–1.43)	0.141
Urbanization					
1 (highest)	40	Ref.		Ref.	
2	34	0.92 (0.58–1.46)	0.728	0.86 (0.55–1.37)	0.536
3	27	1.23 (0.75–2.00)	0.410	1.14 (0.69–1.87)	0.610
4	41	1.49 (0.97–2.31)	0.072	1.07 (0.67–1.71)	0.763
CCI Score					
0	87	Ref.		Ref.	
1	24	2.27 (1.45–3.58)	<0.001	1.24 (0.77–2.01)	0.377
≥2	31	3.96 (2.62–5.98)	<0.001	1.92 (1.19–3.10)	0.007
Baseline Comorbidity					
Diabetes	40	1.68 (1.16–2.42)	0.006	1.14 (0.70–1.86)	0.600
Hypertension	88	2.37 (1.69–3.32)	<0.001	0.91 (0.54–1.55)	0.735
Hyperlipidemia	31	0.97 (0.65–1.45)	0.901	0.68 (0.42–1.11)	0.122
Atrial Fibrillation	1	0.53 (0.07–3.82)	0.531	0.21 (0.03–1.57)	0.129
Valvular heart disease	8	1.16 (0.57–2.37)	0.684	0.74 (0.35–1.54)	0.420
Parkinsonism	2	1.67 (0.41–6.77)	0.470	0.65 (0.16–2.64)	0.543
Autoimmune disease	2	0.80 (0.20–3.22)	0.752	1.02 (0.25–4.17)	0.981

Abbreviations: HR, hazard ratio; CI, confidence interval. Adjusted HR: adjusted for sex, age, and all comorbidities in Cox proportional hazards regression.

**Table 3 medicina-59-01920-t003:** Incidence rates, hazard ratios, and confidence intervals of TB in different stratifications.

Variables	Matched Cohort			AAV Cohort							
	n = 9028			n = 2257			Crude HR	*p*-Value	Adjusted HR	*p*-Value	*P* for Interaction
	Event	Person-Years	IR	Event	Person-Years	IR	(95% CI)		(95% CI)		
Overall	104	66,914	15.54	38	17,385	21.86	1.41 (0.98–2.05)	0.067	1.48 (1.02–2.15)	0.041	
Sex											0.001
Female	28	36,191	7.74	23	9579	24.01	3.10 (1.79–5.39)	<0.001	3.24 (1.85–5.67)	<0.001	
Male	76	30,722	24.74	15	7806	19.22	0.78 (0.45–1.36)	0.385	0.78 (0.45–1.36)	0.384	
Age at Baseline (years)											0.707
<40	4	16,434	2.43	4	4539	8.81	3.60 (0.90–14.40)	0.070	4.21 (1.00–17.79)	0.050	
40–65	31	33,110	9.36	10	8745	11.44	1.23 (0.60–2.50)	0.573	1.18 (0.58–2.42)	0.643	
≥65	69	17,370	39.72	24	4101	58.52	1.47 (0.92–2.34)	0.103	1.44 (0.90–2.30)	0.128	
CCI Score											0.597
0	65	55,435	11.73	22	14,321	15.36	1.31 (0.81–2.13)	0.272	1.43 (0.88–2.33)	0.149	
1	16	6400	25.00	8	1979	40.42	1.62 (0.69–3.79)	0.266	1.74 (0.72–4.23)	0.220	
≥2	23	5079	45.29	8	1085	73.76	1.62 (0.73–3.63)	0.237	1.53 (0.67–3.53)	0.316	
Baseline Comorbidity											
Diabetes											0.514
No	76	54,278	14.00	26	14,202	18.31	1.31 (0.84–2.05)	0.234	1.37 (0.87–2.14)	0.172	
Yes	28	12,636	22.16	12	3183	37.70	1.77 (0.90–3.47)	0.100	1.94 (0.97–3.90)	0.062	
Hypertension											0.435
No	41	39,553	10.37	13	10,602	12.26	1.18 (0.63–2.21)	0.599	1.22 (0.65–2.29)	0.538	
Yes	63	27,361	23.03	25	6783	36.86	1.62 (1.02–2.58)	0.041	1.75 (1.10–2.79)	0.019	
Hyperlipidemia											0.286
No	83	51,970	15.97	28	13,870	20.19	1.27 (0.83–1.95)	0.276	1.33 (0.86–2.04)	0.200	
Yes	21	14,943	14.05	10	3515	28.45	2.06 (0.97–4.37)	0.061	2.59 (1.19–5.60)	0.016	

Abbreviations: IR, incidence rates per 10,000 person-years; HR, hazard ratio; CI, confidence interval. Adjusted HR: adjusted for gender, age, and all comorbidities in Cox proportional hazards regression.

**Table 4 medicina-59-01920-t004:** Incidence rates, hazard ratios, and confidence intervals of TB in different follow-up period.

Follow-Up Period, Year	Matched Cohort			AAV Cohort						
	n = 9076			n = 2269			Crude HR	*p*-Value	Adjusted HR	*p*-Value
	Event	Person-Years	IR	Event	Person-Years	IR	(95% CI)		(95% CI)	
Years of Follow-up										
<2	31	17,415	17.80	14	4355	32.15	1.81 (0.96–3.39)	0.067	1.91 (1.01–3.60)	0.046
2–5	37	21,473	17.23	9	5467	16.46	0.96 (0.46–1.98)	0.900	1.03 (0.49–2.14)	0.941
>5	36	28,027	12.84	15	7564	19.83	1.55 (0.85–2.83)	0.153	1.63 (0.88–2.99)	0.118

## Data Availability

The data presented in this study are available on request from the corresponding author.

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
