# Peer review of "Antineutrophil Cytoplasmic Antibody-Associated Vasculitis and the Risk of Developing Incidental Tuberculosis: A Population-Based Cohort Study"

_medicina, 2023, doi:10.3390/medicina59111920_

Round 1

Reviewer 1 Report

Comments and Suggestions for Authors

The manuscript titled "Antineutrophil Cytoplasmic Antibody-associated Vasculitis and the Risk of Developing Incidental Tuberculosis: A Population-based Cohort Study" presents a study investigating the risk of incidental tuberculosis (TB) in patients diagnosed with antineutrophil cytoplasmic antibody-associated vasculitis (AAV). While the study addresses an important topic, there are several shortcomings and mistakes in various sections of the manuscript:

  1. Abstract:

    • The abstract lacks a concise summary of the study's key findings. It mentions the increased risk of TB in AAV patients but does not provide specific numerical results or the clinical implications of these findings.
    • The abstract should briefly discuss the limitations of the study to provide readers with a balanced understanding of the research.
  2. Introduction:

    • The introduction provides a broad overview of AAV and TB but lacks a clear research question or hypothesis. It should explicitly state the primary objective of the study.
    • The introduction mentions the high risk of infection in AAV patients but doesn't cite specific sources or studies to support this claim. Providing relevant references would strengthen the introduction's credibility.
  3. Materials and Methods:

    • The section describing the study population is unclear and contains unnecessary details. It could be streamlined for clarity.
    • The explanation of the selection criteria for GPA and MPA patients is convoluted and should be simplified.
    • The description of the propensity score matching process lacks clarity and details. Readers need more information about the variables used for matching and the rationale behind them.
    • The section does not mention how missing data were handled, which is essential for understanding potential bias.
  4. Results:
    The interpretation of results should be concise and focused on the main findings. The section appears repetitive, and some details could be moved to the discussion.

  5. Discussion:
    The discussion starts by reiterating the main findings but does not delve into their clinical significance or implications.
    The manuscript mentions a "subgroup effect" without adequately explaining what this means or why it is relevant. It needs further clarification.
    While the discussion discusses potential mechanisms linking AAV and TB, it does not thoroughly evaluate the existing literature on this topic or cite relevant studies.
    The discussion should also address the limitations more comprehensively, including the issues related to data accuracy, misclassification, and overestimation of AAV cases.

  6. Conclusions:
    The conclusions section summarizes the study's findings but does not offer practical recommendations for clinicians or public health practitioners.
    It should emphasize the clinical relevance of the findings and the importance of TB screening and prevention in AAV patients.

Author Response

Responses to comments from Reviewer #1

We would like to thank you for your extensive assessment of our manuscript, and deeply appreciate your valuable comments. We have taken all the remarks into account, in a manner that is described in detail below together with our answers to certain comments. We think that, following the reviewers’ suggestions, our manuscript has gained in clarity and hope that the changes made will be considered satisfactory.

Comment 1:

The abstract lacks a concise summary of the study's key findings. It mentions the increased risk of TB in AAV patients but does not provide specific numerical results or the clinical implications of these findings.

Reply:

Thank you for the suggestions. We have added specific numerical results and clinical implications in the conclusion section. We hope that will be considered satisfactory.

Page 1, line 32-36 (please refer to revised track version)

“3.24;95%CI, 1.85-5.67). Conclusion: Patients with AAV exhibit a 48% elevated TB risk, notably a 91% increase within the first two years post-diagnosis. Female AAV patients face a 3.24 times higher TB risk compared to females without AAV. The study is limited by potential misclassification and overestimation of AAV cases. Clinicians should closely monitor TB risk in AAV patients, especially in females and the initial two years following diagnosis.”

Comment 2:

The abstract should briefly discuss the limitations of the study to provide readers with a balanced understanding of the research.

Reply:

Thank you for the suggestions, We have added limitations of potential misclassification and overestimation of AAV cases in the conclusion section. We hope that will be considered satisfactory.

Page 1, line 34-35 (please refer to revised track version)

“TB risk compared to females without AAV. The study is limited by potential misclassification and overestimation of AAV cases. Clinicians should closely monitor TB risk in AAV patients, especially”

Comment 3:

The introduction provides a broad overview of AAV and TB but lacks a clear research question or hypothesis. It should explicitly state the primary objective of the study.

Reply:

Thank you for the suggestion. Our research question is incidental TB risk following AAV diagnosis. We have revise relevant content in the introduction section and hope that will be considered satisfactory.

Page 2, line 55-58 (please refer to revised track version)

“mined that 42% of the instances of infection were pulmonary in nature [6]. In TB endemic area, understanding the risk of incidental TB following AAV diagnosis is crucial. However, due to the relative low prevalence of AAV and TB, researches exploring AAV and subsequent TB risk are lacking.”

Comment 4:

The introduction mentions the high risk of infection in AAV patients but doesn't cite specific sources or studies to support this claim. Providing relevant references would strengthen the introduction's credibility.

Reply:

Thank you for the suggestion. AAV patients face a high risk of infection. In a 325-patient AAV cohort with 2,307 patient-year of follow-up, 40% had at least one severe infection. The incidence rate was 9.1/100 patient-years, peaking in the first year post-diagnosis [Rheumatology 2021;8;60(6):2745-2754]. In an European Vasculitis Society trials, infection led to the hospitalization of 30% of the AAV patients and was the leading cause of death [Ann Rheum Dis 2011;70:488-94]. Another report on 489 AAV patients determined that 42% of the instances of infection were pulmonary in nature [Nephrol Dial Transplant 2015;30 Suppl 1:i171-81].

We have added relevant content in the introduction section and hope that will be considered satisfactory.

Page 2, line 50-55 (please refer to revised track version)

“AAV patients face a high risk of infection. In a 325-patient AAV cohort with 2,307 patient-year of follow-up, 40% had at least one severe infection. The incidence rate was 9.1/100 patient-years, peaking in the first year post-diagnosis [4]. In an European Vasculitis Society trials, infection led to the hospitalization of 30% of the AAV patients and was the leading cause of death [4]. Another report on 489 AAV patients determined that 42% of the instances of infection were pulmonary in nature”

Comment 5:

The section describing the study population is unclear and contains unnecessary details. It could be streamlined for clarity.

The explanation of the selection criteria for GPA and MPA patients is convoluted and should be simplified.

Reply:

Thank you for the suggestions. We have rewritten the “Study Population” section and the selection criteria for GPA and MPA, striving to minimize redundant phrasing. We hope that will be considered satisfactory.

Page 3, line 94-109, (please refer to revised track version)

“2.3. Selection of patients with ANCA-associated vasculitis

All participants underwent a 12-month burn-in period. Individuals with any of following diagnosis during burn-in period were excluded: (1) GPA or MPA diagnosis; (2) hepatitis B (ICD-9 CM: 070.2, 070.3, and V02.61) or hepatitis C (ICD-9 CM: 070.41, 070.44, 070.51, 070.54, and V02.62); (3) diagnosis of renal failure (ICD-9 CM: 639.3, 586, V56.8, and V45.1), glomerulonephritis (ICD-9 CM: 580-582, and 583.1-583.4); (4) hemoptysis (ICD-9 CM: 786.3).

The identify of GPA cohort was made by individuals aged ≥ 20 years with medical claims of GPA (ICD-9-CM diagnosis code 446.4) at least two outpatient visits or one in-patient visit between January 1, 2000, and December 31, 2012.

The identify of MPA cohort was made by individuals aged ≥ 20 years with medical claims of MPA (ICD-9-CM diagnosis code 447.6) at least two outpatient visits or one in-patient visit between January 1, 2000, and December 31, 2012.

Ultimately, the AAV cohort comprises both the GPA and MPA cohorts. The exclusion criteria were as follows: (1) History of tuberculosis prior to diagnosis of AAV; (2) Missing basic information; (3) Age < 20 years old.”

Comment 6:

The description of the propensity score matching process lacks clarity and details. Readers need more information about the variables used for matching and the rationale behind them.

Reply:

Thank you for the suggestion. We have revised the 'Propensity Score Matching' section to improve both its detail and clarity. We hope that will be considered satisfactory.

Page 3-4, line 138-149, (please refer to revised track version)

“patients and the matched cohort [8]. To further consider the robustness of the correction, we applied 1:4 nearest propensity score matching (PSM) to adjust for the variables listed in Table 1 and compared the results to the multivariate Cox model. PSM via the nearest-neighbor method, initially to the eighth digit and then as required to the first digit, was used in this study. Matches were first made within a caliper width of 0.0000001, and then the caliper width was increased for unmatched cases to 0.1. We reconsidered the matching criteria and performed a rematch according to the greedy algorithm including the following baseline variables: year of index date, age, sex, monthly income, urbanization, CCI score, and comorbidities (diabetes, hypertension, hyperlipidemia, atrial fibrillation, valvular heart disease, parkinsonism, and autoimmune disease). A standardized mean difference of < 0.1 indicated a negligible difference in the distribution between both cohorts [9-10].”

Comment 7:

The section does not mention how missing data were handled, which is essential for understanding potential bias.

Reply:

Thank you for the suggestion. In research using Taiwan's National Health Insurance Research Database, missing data are typically excluded. However, the current study contains no missing data. We have added relevant content in “statistical analysis” section, and hope that will be considered satisfactory.

Page 4-5, line 188-189, (please refer to revised track version)

“subgroups and the risk of incidental TB. In research using Taiwan's NHIRD, missing data are typically excluded. However, the current study contains no missing data. All statistical”

Comment 8:

The interpretation of results should be concise and focused on the main findings. The section appears to be repetitive, and some details could be moved to the discussion

Reply:

Thank you for the suggestion. We have consolidated the content of the original subsection “3.1 AAV Cohort” into “3.2 Baseline Demographic Data”. Additionally, we have streamlined redundant material in the original subsection “3.4 Risk of Incidental TB: Stratification and Interaction Tests” to enhance the reader's focus on our results. We hope that will be considered satisfactory.

Page 5, line 196-197, (please refer to revised track version)

“The study flow is in figure1. Our study population included 2,257 and 9,028 patients in the AAV and matched-cohorts, respectively.”

Page 9, line 233-244, (please refer to revised track version)

“3.3. Risk of incidental TB: Stratification and interaction tests

As shown in Table 3, the risk of developing incidental TB was 3.24 times higher among female AAV patients than among females in the matched cohort (adjusted HR: 3.24, 95% CI = 1.85 – 5.67), with significant p value for interaction (p for interaction =0.001). The risk of contracting incidental TB was similar between the two groups when stratified in terms of age, CCI score, diabetes, hypertension, or hyperlipidemia. There was no significant interaction by age, CCI score, diabetes, hypertension, or hyperlipidemia. (age: p for interaction = 0.707; CCI score: p for interaction = 0.597; diabetes: p for interaction = 0.514; hypertension: p for interaction = 0.435; hyperlipidemia: p for interaction = 0.286). As shown in Table 4, AAV patients who”

Comment 9:

The discussion starts by reiterating the main findings but does not delve into their clinical significance or implications.

Reply:

Thank you for the suggestion. We have streamlined redundant content in this paragraph and articulated the clinical implications of this study, specifically recommending that clinicians should closely monitor TB risk in AAV patients, with particular attention to females and the first two years following diagnosis. We hope that will be considered satisfactory.

Page 10, line 254-265, (please refer to revised track version)

“This nationwide population-based study using propensity score matching provided strong evidence that the risk of contracting incidental TB was nearly 50% higher in AAV patients than those without AAV. Female AAV patients had nearly 3-fold higher risk of subsequent TB than females without AAV. When compared with matched cohort, the risk of incidental TB was highest for the first 2 years after diagnosis of AAV, with nearly 1-fold increased risk. Clinicians should closely monitor subsequent TB risk in AAV patients, especially in females and the initial two years following diagnosis”

Comment 10:

The manuscript mentions a "subgroup effect" without adequately explaining what this means or why it is relevant.

Reply:

In the context of this study, if a significant "p for interaction" is observed between gender and AAV, this would indicate that the impact of AAV varies significantly between male and female subjects. To mitigate potential confusion, we have revised the sections of the manuscript that mention 'subgroup effect' to more accurately interpret this interaction. We hope that will be considered satisfactory.

Page 9, line 236-241, (please refer to revised track version)

“3.24, 95% CI = 1.85 – 5.67), with significant p value for interaction (p for interaction =0.001). The risk of contracting incidental TB was similar between the two groups when stratified in terms of age, CCI score, diabetes, hypertension, or hyperlipidemia. There was no significant interaction by age, CCI score, diabetes, hypertension, or hyperlipidemia.”

Page 10, line 257-258, (please refer to revised track version)

Female AAV patients had nearly 3-fold higher risk of subsequent TB than females without AAV.”

Comment 11:

While the discussion discusses potential mechanisms linking AAV and TB, it does not thoroughly evaluate the existing literature on this topic or cite relevant studies.

Reply:

Current research predominantly focuses on the potential of TB to trigger AAV or on the prevalence of P-ANCA and C-ANCA in TB patients. However, there is a paucity of studies investigating the risk of subsequent incidental TB following a diagnosis of AAV. The increased susceptibility of AAV patients to TB could be attributed to T-cell response dysregulation and the immunosuppressive medications prescribed for AAV management. We refer to two studies to elucidate the mechanisms underlying T-cell response dysregulation in AAV patients [Initiation and regulation of T-cell responses in tuberculosis. Mucosal Immunol 2011;4:288-93], [Imbalance of Circulatory T Follicular Helper and T Follicular Regulatory Cells in Patients with ANCA-Associated Vasculitis. Mediators Inflamm 2019;8421479]. One mechanism implicates delayed induction of TB-specific, Foxp3+ regulatory T cells (Treg). AAV patients have demonstrated elevated levels of circulatory T follicular helper (Tfh) and T follicular regulatory (Tfr) cells, along with an increased Tfh2/Tfh1 ratio. It is conceivable that this Treg imbalance could contribute to TB development in AAV patients.

Additionally, we cite two studies highlighting the potential impact of immunosuppressants on the risk of TB in AAV patients [Risk of incidental active tuberculosis and use of corticosteroids. Int J Tuberc Lung Dis 2015;19:936-42], [Increased risk of mycobacterial infections associated with anti-rheumatic medications. Thorax 2015;70:677-82]. A population-based nested case-control study in Taiwan involving nearly 6,000 TB patients revealed that all categories of corticosteroid use—current, recent, past, ever, and chronic—were linked to a heightened risk of incidental TB. Similarly, a Canadian population-based nested case-control study indicated that the current use of disease-modifying anti-rheumatic drugs, including cyclophosphamide, azathioprine, and cyclosporine, was correlated with an elevated risk of developing incidental TB, with an adjusted odds ratio of 23

Page 10, line 268-271, (please refer to revised track version)

“3.60) within the first 2 years after diagnosis of AAV. Current research predominantly focuses on the potential of TB to trigger AAV or on the prevalence of P-ANCA and C-ANCA in TB patients [11-12]. However, there is a paucity of studies investigating the risk of subsequent incidental TB following a diagnosis of AAV. AAV is an inflammatory disease”

Comment 12:

The discussion should also address the limitations more comprehensively, including the issues related to data accuracy, misclassification, and overestimation of AAV cases.

Reply:

Thank you for the suggestion. We have revised relevant content about limitation of data accuracy, misclassification, and overestimation of AAV cases. Regarding data accuracy, the algorithm used to identify AAV cases has not been validated, posing a potential risk for data accuracy bias.

As for the risk of misclassification, we have previously elaborated on the limitations imposed by the ICD-9-CM coding system. Our diagnostic criteria for AAV amalgamate various guidelines—namely the American College of Rheumatology (ACR), the Chapel Hill Consensus Conference (CHCC), and The European Medicines Agency (EMA)—none of which clearly distinguish between GPA and MPA. Although these criteria complexities introduce a risk of misclassification, it is mitigated by our strategy of collectively analyzing GPA and MPA as a unified AAV group.

Concerning the overestimation of AAV cases, our study reports an incidence higher than commonly observed. To rationalize this discrepancy, we referenced a Taiwanese longitudinal, retrospective study on kidney diseases [2019 Annual Report on Kidney Disease in Taiwan]. This study, covering the period from January 2015 to December 2019, reported that 4.1% of primary glomerulonephritis cases involved AAV. Our study, based on the NHIRD and spanning 2000 to 2012, encompassed 53,839 glomerulonephritis cases. Applying the 4.1% rate yields an estimated 2,154 AAV cases, a figure remarkably consistent with the actual 2,257 cases we identified. This comparative exercise lends credibility to our elevated AAV incidence, although it also underscores the potential for overestimation. Overcoming this limitation will require large-scale, multi-center, randomized-controlled trials. We hope that will be considered satisfactory.

Page 12, line 349-350, (please refer to revised track version)

“trative claims-oriented databases. Fourth, the algorithm employed for AAV case identification has not undergone validation, potentially introducing bias related to data accuracy.”

Comment 13:

The conclusions section summarizes the study's findings but does not offer practical recommendations for clinicians or public health practitioners. It should emphasize the clinical relevance of the findings and the importance of TB screening and prevention in AAV patients.

Reply:

Thank you for the suggestion. We have re-written “conclusions” section by your suggestion, and hope that will be considered satisfactory.

Page 12, line 355-358, (please refer to revised track version)

“Patients with AAV exhibit a 48% elevated TB risk, notably a 91% increase within the first two years post-diagnosis. Female AAV patients face a 3.24 times higher TB risk compared to females without AAV. Clinicians should closely monitor TB risk in AAV patients, especially in females and the initial two years following diagnosis”

Reviewer 2 Report

Comments and Suggestions for Authors

This is a very interesting paper examining the association between ANCA-associated vasculitis and the development of tuberculosis.

Some modifications would further interest the reader.

The revisions are presented below.

Abstract

No modifications are necessary.

Introduction

The purpose of this study should be clearly stated at the end of the Introduction.

Materials and Methods

Indicate whether written consent will be provided to participants and how they can opt out of the study.

Discussion

A limitation of this study is that actual drug use, which is likely to be associated with the development of TB, was not investigated. A mention of this and a description of future developmental studies is needed.

Author Response

Responses to comments from Reviewer #2

We would like to thank you for your extensive assessment of our manuscript, and deeply appreciate your valuable comments. We have taken all the remarks into account, in a manner that is described in detail below together with our answers to certain comments. Due to word count limit, there were partial contents in the supplementary files. We think that, following the reviewers’ suggestions, our manuscript has gained in clarity and hope that the changes made will be considered satisfactory.

Comment 1:

The purpose of this study should be clearly stated at the end of the Introduction.

Reply:

        Thank you for the suggestion. Our study aims to assess the risk of incidental TB following AAV diagnosis. We have revised relevant content at the end of the introduction and hope that will be considered satisfactory.

Page 2, line 61-64 (please refer to revised track version)

“Our study aims to assess the risk of incidental TB following AAV diagnosis. Using a novel algorithm [7], we identified patients with granulomatosis with polyangiitis (GPA) and microscopic polyangiitis (MPA) from a national database to evaluate their subsequent TB risk.”

Comment 2:

Indicate whether written consent will be provided to participants and how they can opt out of the study.

Reply:

Thank you for the suggestion. Patient consent was waived due to the retrospective nature of this study, and Participant data will be de-identified to ensure anonymity. We hope that will be considered satisfactory. We have added relevant content in “materials and methods” section, and hope that will be considered satisfactory.

Page 2, line 82-84 (please refer to revised track version)

“Classification of Disease, Ninth Revision, Clinical Modification (ICD-9-CM). Patient consent was waived due to the retrospective nature of this study, and Participant data will be de-identified to ensure anonymity. The Research Ethics Committee of China Medical”

Comment 3:

A limitation of this study is that actual drug use, which is likely to be associated with the development of TB, was not investigated. A mention of this and a description of future developmental studies is needed.

Reply:

Thank you for the suggestion. We have added relevant content in our “limitation” section, and hope that will be considered satisfactory.

Page 12, line 351-353 (please refer to revised track version)

“Fifth, this study does not account for the impact of immunosuppressive agents commonly used in AAV patients, such as steroids, cyclophosphamide, rituximab, azathioprine, or cyclosporine on the risk of subsequent TB. Future developmental studies are needed.”

Round 2

Reviewer 1 Report

Comments and Suggestions for Authors

The manuscript has been significantly improved.